# Compressed Gaussian Estimation under Low Precision Numerical Representation

**DOI:** 10.3390/s23146406

**Published:** 2023-07-14

**Authors:** Jose Guivant, Karan Narula, Jonghyuk Kim, Xuesong Li, Subhan Khan

**Affiliations:** 1School of Mechanical and Manufacturing Engineering, University of New South Wales, Sydney, NSW 2052, Australia; 2Independent Researcher, Bangkok 10100, Thailand; karan_819@hotmail.com; 3Naif Arab University for Security Sciences, Riyadh 14812, Saudi Arabia; jkim@nauss.edu.sa; 4College of Science, Australia National University, Canberra, ACT 2601, Australia; xuesong.li@anu.edu.au; 5School of Electrical and Information Engineering, University of Sydney, Camperdown, NSW 2006, Australia

**Keywords:** CEKF, compressed Kalman filter, compressed estimation, high dimensional estimation, low precision numerical format, integer precision covariance

## Abstract

This paper introduces a novel method for computationally efficient Gaussian estimation of high-dimensional problems such as Simultaneous Localization and Mapping (SLAM) processes and for treating certain Stochastic Partial Differential Equations (SPDEs). The authors have presented the Generalized Compressed Kalman Filter (GCKF) framework to reduce the computational complexity of the filters by partitioning the state vector into local and global and compressing the global state updates. The compressed state update, however, still suffers from high computational costs, making it challenging to implement on embedded processors. We propose a low-precision numerical representation for the global filter, such as 16-bit integer or 32-bit single-precision formats for the global covariance matrix, instead of the expensive double-precision, floating-point representation (64 bits). This truncation can inevitably cause filter instability since the truncated covariance matrix becomes overoptimistic or even turns to be an invalid covariance matrix. We introduce a Minimal Covariance Inflation (MCI) method to make the filter consistent while minimizing the truncation errors. Simulation-based experiments results show significant improvement of the proposed method with a reduction in the processing time with minimal loss of accuracy.

## 1. Introduction

The Bayesian estimation of the state of high dimensional systems is a relevant topic in diverse research and application areas. When a Gaussian estimator is applied in an optimal way, the full covariance matrix needs to be maintained during the estimation process; for that, the numerical format commonly used for representing the elements of the covariance matrix is the double precision floating point (e.g., the 64 bits IEEE-754) or, in some cases, in which the covariance matrix characteristic allows it, the single precision floating point (32 bits IEEE-754). However, in certain cases, single precision can result in numerical problems. Numerical formats of even lower precision, e.g., 16 bits, are usually more difficult to be applied, due to usual numerical instability, lack of consistency or, in the best cases where the approximation is properly applied (by performing matrix inflation), highly conservative results. However, when properly treated, a full Gaussian filter can operate in low precision without incurring relevant errors or excessive conservativeness. In particular, for systems that can be processed through a Generalized Compressed Kalman Filter (GCKF) [1], it is possible to operate under a lower precision numerical format. The advantage of maintaining and storing the huge covariance matrix using low precision is relevant because the required memory can be reduced to a fraction of the nominal required amount. In addition to reducing the amount of required data memory, the fact that a program operates using lower amount of memory, improves its performance by lowering the number of RAM cache misses, a limitation which is still present in the current computer technology. The frequent RAM cache misses lead to additional problems such as high memory traffic, energy consumption, and additional issues that decrease the efficiency of CPU utilization.

This effect is even more relevant when those operations do occur massively and at high frequency. That is the case of a standard Gaussian estimator performing update steps, because those require updating the full covariance matrix, which means the full covariance matrix is read and written at each KF update, i.e., all the elements of the covariance matrix are R/W acceded. If, in addition to the high dimensionality and high frequency operation, the estimator also includes the capability of performing smoothing, then the requirements of memory can be dramatically increased, further exacerbating the problem.

The approach presented in this paper exploits the low-frequency nature of the global updates of the GCKF in combination with a simple matrix bounding technique, for treating truncation errors associated with low precision numerical representation. In the GCKF framework, only during the low-frequency global updates the full covariance matrix is modified. That particularity of the GCKF allows performing additional processing on the full covariance matrix, if required for diverse purposes. As the global updates are performed at a low execution rate, the cost of the treatment of the full covariance matrix is amortized over the overall operation, in many cases making the overhead result in very low extra cost. One of the purposes of treating the full covariance matrix, P, can be for approximating it by a low precision version of it; e.g., for representing the covariance matrix in single precision format or even by lower precision representations such as scaled 16-bit integers. Just truncating precision (truncating bits) is not adequate for replacing a covariance matrix; consequently, proper relaxation of the matrix needs to be performed. One way of doing it is by generating a bounding matrix, $P^{*}$, which satisfies two conditions:The approximating matrix, $P^{*}$, must be more conservative than the original one (the one being approximated), i.e., $P^{*}-P$ must be positive semidefinite; this fact is usually expressed as $P^{*}-P\geq 0$ or as $P^{*}\geq P$.$P^{*}$ is represented through a lower precision numerical format.

In addition, it is desirable that the discrepancy $P^{*}-P$ should be as small as possible to avoid over-conservative estimates, particularly if the approximation needs to be applied repeatedly. This technique would allow a GCKF engine to operate servicing a number of clients, that when a request to perform a global update is received, the required memory for temporary storing and processing the full covariance matrix is partially or fully provided by the engine, while the memory for storing the low precision version of the full covariance matrixes is maintained by the clients, which means that the actual required memory for double precision is only one and it is shared by many client processes. Furthermore, by proper manipulation of the required operations in the prediction and update steps, many of those do not need to be performed in full double precision, but just requiring high precision temporary accumulators of low memory footprint.

In addition, the required precision can be dynamically decided according to the availability of memory resources, observability of the estimation process, and required accuracy of the estimates. The technique would also allow maintaining copies of the full covariance matrix at different times (regressors of matrixes), e.g., for cases of smoothing.

## 2. The Generalized Compressed Kalman Filter (GCKF)

The GCKF is the approach introduced in [1]. The approach is intended to process estimation problems in high dimensional cases, in which the state vector is composed of hundreds or thousands of scalar components. Those estimation processes may need to operate at high processing rates for systems whose dynamics are fast and high dimensional, such as certain Stochastic Partial Differential Equations (SPDEs), and also for certain Simultaneous Localization and Mapping (SLAM) applications.

The GCKF divides the estimation process into a low-frequency global component that updates a high dimensional Gaussian Probability Density Function (PDF) at a lower rate. In addition to that, the estimation process maintains several lower dimensional estimation processes (Individual estimations processes, IEPs), which are operated at a high processing rate, to deal with the fast dynamics of the system. The approach has been exploited in centralized multi-agent SLAM (as shown in [1]) and in treating SPDE (such as in [2] and in [3]). A precursor of the GCKF, known as Compressed Extended Kalman Filter (CEKF) had been used mostly in mono-agent SLAM, such as in [4,5], and in subsequent work usually for localization of aerial platforms [6]. Some variants of the CEKF and of the GCKF have been adapted to exploit other cores than the usual EKF, such as Unscented Kalman Filter (UKF) and Cubature Kalman Filter (CuKF) based cores, which are usually better suited for certain non-linear cases [3,6].

## 3. Other Well-Known Approaches

Other approaches exist in the literature for reducing the computational cost of the KF-based methods which is a necessity in high dimensional problems especially when real-time estimates are a requirement or when computational resources are limited such as on embedded systems. These can be broadly classified into three categories: (i) error subspace KFs, (ii) distributed/decentralization methods, and (iii) methods rooted in optimizations that exploit the characteristics of the estimation problems [7].

The error subspace KFs reduce the computational cost by using a lower rank approximation of the covariance matrix. The methods under this category include the reduced rank square root algorithm [8], the singular evolutive extended Kalman (SEEK) filter [9], and the Ensemble KF (EnKF) [10] which instead use an ensemble of states to represent the error statistics instead of the mean vector and the covariance matrix in EKF. Although the EnKF is the preferred method when dealing with high dimensional estimation problems, it often requires a large number of observations for convergence in estimating strongly non-linear problems [11]. Furthermore, the ensemble representation is ideal for dealing with cases of strong correlation but that is not a general characteristic of all the estimation problems [2]. The optimizations employed in the EnKF are also based on conservative assumptions. The local analysis method, for example, assumes that the statistical dependency is localized [12] to reduce the computational cost when processing a large number of observations in the update/analysis step. This assumption is valid for the systems modeled by certain PDEs but again it is not a general characteristic of all the estimation problems [2].

The distributed or decentralization methods reduce the computational cost by dividing the estimation problem into a set of sub-problems. The examples of the methods under this classification include: (i) the distributed KF for sensor networks using consensus algorithms in [13,14] where the estimates have been shown to converge to those of the centralized KF with the cost of temporarily sacrificing optimality [15]; (ii) the fully decentralized KF suitable for systems that are sparse and localized such as those resulting from spatio-temporal discretization of PDEs [16]; (iii) the aforementioned GCKF [1], which is not fully decentralized requiring low-frequency global updates, shown to optimally treat problems that allow the estimation process to be divided, during certain periods of time, into a set of lower dimensional sub-problems; and, (iv) GCKF with switching technique and architecture for information exchange between sub-processes [2,7] targeting previously intractable cases, i.e., problems that can’t be decoupled into lower dimensional sub-problems.

There is a large literature that has exploited various characteristics and structures of the estimation problem to reduce the computational cost of KF and its variants. Some notable examples include: [17,18] when the process or the observation model is partly linear, [17,19,20] when the measurements are conditionally linear, [21] when only part of the state is observed through the measurement model, [4,6] when the system’s structure contains unobserved and static states consistently for a period of time, [1] when the system’s process and observation models can be decoupled into a set of sub-processes, [17] when the number of measurements is large and the measurement noise covariance matrix is diagonal or block-diagonal, and [17] when the number of observations is greater than the dimensionality of the states.

The use of a lower precision numerical format in high dimensional problems such as that employed in this paper is previously observed in EnKF [22,23]. However, no matrix inflation or matrix bounding techniques were applied as they relied on the assumption that the numerical errors were within the tolerance of the system. They instead redistributed the saved computational resources from lower precision arithmetic and lower memory usage to employing a larger ensemble size which provided a better estimate of the underlying distribution.

## 4. Low-Precision Approximating Covariance Matrix

In general, for any covariance matrix, $P=\{p_{i,j}\}$, its non-diagonal elements can be expressed as follows:(1)$$\begin{array}{cc} p_{i,j} & =\sqrt{p_{i,i}\cdot p_{j,j}}\cdot \phi_{i,j}\\ \phi_{i,j} & \in [-1,1] \end{array}$$

This fact can be expressed in matrix terms (for the purpose of explaining it, but not for implementation reasons),
(2)$$\begin{array}{cc} P & =D\cdot \Phi \cdot D\\ d_{i,j} & =\left\{\begin{array}{cc} \sqrt{p_{i,i}} & \forall i=j\\ 0 & \forall i\neq j \end{array}\right.\\ \phi_{i,j} & =\frac{p_{i,j}}{\sqrt{p_{i,i}\cdot p_{j,j}}}\\ \phi_{i,j} & \in [-1,+1]\\ \phi_{i,i} & =1,\;\;\forall i \end{array}$$
where the full covariance matrix is expressed by a scaling matrix D (which is diagonal) and a full dense symmetric matrix Φ whose elements are bounded in the range [−1,+1]. The matrix Φ is in fact a normalized covariance matrix. The low precision representation, introduced in this work, keeps the diagonal matrix D (i.e., its diagonal elements) in its standard precision representation (e.g., single, double, or long double precision), and it approximates the usually dense matrix Φ by a low precision integer numerical format. For instance, if a precision of *N* bits is required, then Φ is fully represented by the integer matrix μ whose elements are signed integers of *N* bits. For obtaining the elements of μ, the relation is as follows,
(3)$$\begin{array}{ccc} \mu_{i,j}=f\left(\phi_{i,j}\right) & = & \left[2^{N-1}\cdot \phi_{i,j}\right]\\ -1\leq \phi_{i,j}\leq 1 & \Rightarrow & -2^{N-1}\leq \mu_{i,j}\leq 2^{N-1} \end{array}$$

The operator [·] means that the argument is approximated to the nearest integer (i.e., “rounded”). The inverse operation produces a discrete set of elements in the range of real numbers [−1,+1], and it is defined as follows,
(4)$$\begin{array}{ccc} \phi_{i,j}^{*} & = & \mu_{i,j}\cdot \frac{1}{2^{N-1}}\\ -2^{N-1}\leq \mu_{i,j}\leq 2^{N-1} & \Rightarrow & -1\leq \phi_{i,j}^{*}\leq 1 \end{array}$$

The discrepancy between Φ and its low precision version, $\Phi^{*}$, is simply the matrix residual $\Phi -\Phi^{*}$. This discrepancy matrix can be bounded by a diagonal matrix, B, that must satisfy the condition $B+\Phi^{*}-\Phi >0$. A “conservative” bounding diagonal matrix, is as it was introduced in [5,21], or adapted for this case, as follows,
(5)$$b_{i,j}=\left\{\begin{array}{cc} 0 & \forall i\neq j\\ \sum_{\begin{array}{c} k=1\\ k\neq i \end{array}}^{n}\left|\phi_{i,k}^{*}-\phi_{i,k}\right| & \forall i=j \end{array}\right.$$

This type of approximation is deterministic, i.e., always valid, and it may be too conservative for cases such as in this application, for dealing with limited numerical precision. The resulting diagonal elements are deterministically calculated; however, they can be simply estimated by considering that the added values do follow a uniform distribution, as usually is the case of the rounding errors. For the case of adding a high number of independently and identically distributed random variables, whose PDF is uniform, the resulting PDF is almost Gaussian. As the rounding errors follow a uniform distribution, a less conservative and “highly probable” bounding matrix for the error matrix can be simply defined as follows,
(6)$$b_{i,j}=\left\{\begin{array}{cc} 0 & \forall i\neq j\\ \frac{5}{4}\cdot \sqrt{n}\cdot \frac{1}{2^{N}} & \forall i=j \end{array}\right.$$

In which *n* is the size of the matrix (more exactly P is a square matrix of size *n* by *n*), and *N* is the number of bits of the integer format being used. The bound $1.25\cdot \sqrt{n}\cdot 2^{-N}$ is usually much lower than the conservative bound $\sum_{\begin{array}{c} k=1\\ k\neq i \end{array}}^{n}\left|\phi_{i,k}^{*}-\phi_{i,k}\right|$. This less conservative bound is valid for the cases where the residuals $\phi_{i,k}^{*}-\phi_{i,k}$ follow a uniform distribution in the range $\left[-2^{-N},+2^{-N}\right]$, as it is the case of the truncation error in this approximation. Because of that, an approximating covariance matrix $P^{*}$ is obtained as follows,
(7)$$\begin{array}{cc} P & =D\cdot \Phi \cdot D\\ \; & =D\cdot (\Phi^{*}+\Phi -\Phi^{*})\cdot D\\ \; & =D\cdot \Phi^{*}\cdot D+D\cdot B\cdot D-D\cdot \left(B+\Phi^{*}-\Phi \right)\cdot D<D\cdot \Phi^{*}\cdot D+D\cdot B\cdot D \end{array}$$

This means that the slightly conservative bound for P is given by the matrix $D\cdot \Phi^{*}\cdot D+D\cdot B\cdot D$ in which $\Phi^{*}$ is the reduced precision version of Φ, and the rest of the (implicit) matrixes are high precision (e.g., double) but diagonal. Consequently, the approximating covariance matrix can be stored by using *n* double precision elements and $\frac{n\cdot (n-1)}{2}$ integers. When implemented, the matrix relation is simply evaluated by relaxing the diagonal elements of the nominal covariance matrix, as follows:(8)$$\begin{array}{cc} p_{i,j}^{*} & =\frac{\mu_{i,j}}{2^{N-1}}\cdot \sqrt{p_{i,i}}\cdot \sqrt{p_{j,j}}=\frac{\mu_{i,j}}{2^{N-1}}\cdot \sigma_{i}\cdot \sigma_{j}\\ \mu_{i,j} & =\left[\frac{p_{i,j}}{\sigma_{i}\cdot \sigma_{j}}\cdot 2^{N-1}\right]\;\;\forall i\neq j\\ p_{i,i}^{*} & =p_{i,i}\cdot \left(1+c\right)\\ c & =1.25\cdot \sqrt{n}\cdot \frac{1}{2^{N}}\ll 1 \end{array}$$

In which ${\left\{\sigma \right\}}_{i=1}^{n}$ are simply the standard deviations of the individual state estimates. The apparently expensive conversion defined in Equation (Equation 8), is implemented in a straightforward way, due to the fact that the elements $\phi_{i,j}$ are always bounded in the range [−1,+1]; not even needing any expensive CPU operation, but just reading the mantissas from memory. Only *n* square roots and only *n* divisions need to be evaluated. The rest of the massive operations are additions and products. Alternative integer approximations are also possible. Similarly, the integer approximation could also be implemented via the CEIL truncation operation. Depending on the CPU type and its settings, some of them may be more appropriate than others. The CEIL case is expressed as follows,
(9)$$\begin{array}{ccc} \mu_{i,j} & = & \left\lceil 2^{N-1}\cdot \phi_{i,j}\right\rceil \\ -1\leq \phi_{i,j}\leq 1 & \Rightarrow & -2^{N-1}\leq \mu_{i,j}\leq 2^{N-1} \end{array}$$

In this case, the approximation error $\phi_{i,k}^{*}-\phi_{i,k}$ follows a uniform distribution in the range $\left[0,+2^{-(N-1)}\right]$. It could be shown that the discrepancy is always bounded as expressed in Equation (Equation 7). The resulting required inflation of the diagonal elements, for all the cases (round and CEIL), is marginal provided that the relation $\sqrt{n}\ll 2^{N}$ is satisfied. For instance, if 16 bits are used for the integer approximation, for a 1000×1000 covariance matrix, the required inflation of the diagonal elements results to be
(10)$$\begin{array}{c} p_{i,i}^{*}=p_{i,i}\cdot \left(1+c\right)\\ c=1.25\cdot \sqrt{1000}\cdot \frac{1}{2^{16}}\leq 0.00061=0.061\% \end{array}$$

This means that if the bounding were applied 100 times, the cumulated error would be ${\left(1+c\right)}^{100}$, which for such a small value of *c* would result in a variation of about 100·c, i.e., 6.1%. If 12 bits are used, then the required inflation will be higher; however, it would still be small,
(11)$$\begin{array}{c} p_{i,i}^{*}=p_{i,i}\cdot \left(1+c\right)\\ c=1.25\cdot \sqrt{1000}\cdot \frac{1}{2^{12}}\leq 0.092\% \end{array}$$

An aggressive discretization, e.g., N=8 bits, would require a more evident inflation,
(12)$$c=1.25\cdot \sqrt{1000}\cdot \frac{1}{2^{8}}≃0.155=15.5\%$$

The required low inflation, for the case of n=1000 and N=16, may create the impression that it could be even applied at high frequency rates, e.g., in a standard Gaussian filter. This impression is usually wrong, in particular if the prediction and update steps are applied at very high rates, which would result in a highly inflated covariance matrix. The advantage of using this decorrelation approach, in combination with the GCKF, is that while the estimator operates in a compressed way, the low dimensional individual estimators can permanently operate in high precision (e.g., in double precision) and only at the global updates the precision reduction is applied; this results in marginal inflation of the diagonal elements of the covariance matrix. If more bits can be dedicated for storing P (via the integer matrix), highly tight bounds can be achieved. For instance, in the case of n=1000 if the integer length is N=20 bits,
(13)$$c=1.25\cdot \sqrt{1000}\cdot \frac{1}{2^{20}}≃3.8e^{-5}=0.0038\%$$

For the case of using the GCKF, between 10 and 16 bits seems an appropriate precision, due to the usually low frequency nature of the global updates. If the approximations were to be applied at high frequencies (e.g., not being part of a GCKF process), higher precision formats may be necessary, e.g., N=20. From the previous analysis, it can be inferred that the cumulative approximation error is linear with respect to the rate of application (this is concluded without considering the compensating effect of the information provided by the updates). This is an intuitive estimation because bounding approximations are immersed and interleaved between the prediction and the update steps of the Gaussian estimation process. A more natural way of expressing the same equations would be based in “kilo-states”,
(14)$$c=40\cdot \frac{1}{2^{N}}\cdot \sqrt{K}$$
where *K* is the number of kilo-states, for expressing the dimension of the state vector in thousands of states; and where the constant 40 is simply an easy bound for $1.25\cdot \sqrt{1000}$. The bounding factor of 1.25, which was obtained heuristically, is a conservative value. In the Figure 1, a high number of cases, with high values of *n*, were processed. In each individual case, a covariance matrix was generated randomly, and its bound was evaluated by an optimization process. All the obtained values were lower than 1.25.

The most relevant operation in which the full covariance matrix is fully updated does occur at low processing rate, in the GCKF global update (GU). Still, we process it achieving high precision in the intermediate steps while still using it in its low precision storage. The fact that the H matrix is usually sparse, and of a narrow rectangular shape of size *m* by *n*, with n≫m, implies relevant benefits in the way we can exploit this approach. The H matrix is sparse in the usual cases, and it is also sparse in the virtual update associated with the GU of the GCKF. We can adapt the approach to different modalities of performing a Bayesian update for a Gaussian prior PDF and a Gaussian likelihood function; we give details using the standard KF update equations, as it is one of the ways to perform it. In this section, we explain how to do it for the well-known standard KF update. We focus our analysis on obtaining the posterior covariance matrix, which we call $P^{\left(+\right)}$, while that of the prior PDF is simply indicated as P.
(15)$$\begin{array}{ccc} S & = & H\cdot P\cdot H^{T}+R\\ \; & \vdots & \\ P^{\left(+\right)} & = & P-P\cdot H^{T}\cdot S^{-1}\cdot H\cdot P \end{array}\;$$

We exploit the fact that the observation function is a function of just a subset of the states of the full state vector being estimated. We address those by using integer indexes (which we call here ii), to select only those columns of H which do contain at least one element different to zero. We here express the previous operations using Matlab notation (by which we can clearly indicate the indexing procedure).
(16)$$\begin{array}{cc} S & =H\cdot P\cdot H^{T}+R=H\left(:,\mathrm{ii}\right)\cdot P\left(\mathrm{ii},\mathrm{ii}\right)\cdot H{\left(:,\mathrm{ii}\right)}^{T}+R\\ P^{\left(+\right)} & =P-P\cdot H^{T}\cdot S^{-1}\cdot H\cdot P\\ \Delta P & =P\cdot H^{T}\cdot S^{-1}\cdot H\cdot P\\ H\cdot P & =H(:,\mathrm{ii})\cdot P(\mathrm{ii},:) \end{array}\;$$

This indicates that only a part of the prior matrix P is needed for calculating the variation of P. It means that the intermediate calculations, related to ΔP can be performed incrementally and just requires converting those parts of P to double (or even long double precision, if we wanted). After those parts of ΔP are calculated in high precision we can immediately apply the update to the stored P matrix, which is in single precision, never needing to be entirely converted to double or long double precision. In this way, we obtain the benefits of achieving the accuracy of double (or long double) precision but operating and storing P in low precision, not requiring extra processing, but consuming lower processing time due to the better performance of the CPU (or GPU) memory cache. It is worth noting that ΔP is never fully stored in memory (neither in low nor high precision). The intermediate calculations (usually massive linear combinations) are guaranteed to be performed in high precision (double or even long double), so that numerical errors are minimized, in the same way it would happen if we used double (or long double) representation in the overall process.
(17)$$\begin{array}{cc} H_{\mathrm{ii}} & =double(H(:,\mathrm{ii}))\\ \beta & =double(P(:,\mathrm{ii}))\\ S & =H\left(:,\mathrm{ii}\right)\cdot P\left(\mathrm{ii},\mathrm{ii}\right)\cdot H{\left(:,\mathrm{ii}\right)}^{T}+R=H_{\mathrm{ii}}\cdot P\left(\mathrm{ii},\mathrm{ii}\right)\cdot {H_{\mathrm{ii}}}^{T}+R \end{array}$$
(18)$$\begin{array}{cc} C & =cholesky(S)\\ C_{i} & =C^{-1}\\ S & =C^{T}\cdot C\Rightarrow S^{-1}=C_{i}\cdot {C_{i}}^{T}\\ \gamma & =P\left(:,\mathrm{ii}\right)\cdot {H_{\mathrm{ii}}}^{T}\cdot C_{i}=P\left(:,\mathrm{ii}\right)\cdot \left({H_{\mathrm{ii}}}^{T}\cdot C_{i}\right)\in R^{n\times m} \end{array}$$
(19)$$\begin{array}{cc} K & =P\cdot H^{T}\cdot S^{-1}=\gamma \cdot {C_{i}}^{T}\\ P^{\left(+\right)} & =P-P\cdot H^{T}\cdot S^{-1}\cdot H\cdot P=P-\gamma \cdot \gamma^{T}\\ \Delta P & =\gamma \cdot \gamma^{T} \end{array}\;$$ By exploiting those indexed operations, the nominal cost of updating the covariance matrix is $m\cdot n^{2}+length\left(\mathrm{ii}\right)\cdot m\cdot n$.

The only required temporary matrix to be briefly kept in double precision is γ, which is *n* by *m* (with m≪n).

We can operate on the matrix Φ,
(20)$$\begin{array}{cc} \gamma & =P\left(:,\mathrm{ii}\right)\cdot {H_{\mathrm{ii}}}^{T}\cdot C_{i}=P\left(:,\mathrm{ii}\right)\cdot \left({H_{\mathrm{ii}}}^{T}\cdot C_{i}\right)\\ \; & =D\cdot \Phi \left(:,\mathrm{ii}\right)\cdot D\left(\mathrm{ii},\mathrm{ii}\right)\cdot \left({H_{\mathrm{ii}}}^{T}\cdot C_{i}\right)=D\cdot \mu \\ \mu & =\Phi \left(:,\mathrm{ii}\right)\cdot (D\left(\mathrm{ii},\mathrm{ii}\right)\cdot {H_{\mathrm{ii}}}^{T}\cdot C_{i}) \end{array}\;$$

If, in the estimator update, we operate in terms of the normalized covariance matrix,
(21)$$\begin{array}{cc} \Delta \Phi & =\mu \cdot \mu^{T}\\ \mu & \in R^{n\times length(\mathrm{ii})}\\ \Phi^{+} & =\Phi -\Delta \Phi \\ \phi^{+}\left(i,k\right) & =\phi (i,k)-\langle \mu (:,i),\mu (:,k)\rangle \end{array}\;$$

The inner product 〈μ(:,i),μ(:,k)〉 should be done through a high precision accumulator, even being the elements of μ represented in low precision (the same has been assumed in other operations involving linear combinations of low precision items, in the previous calculations). After this operation, the updated normalized covariance matrix $\Phi^{+}$ is usually not normalized anymore. That fact does not mean, anyway, that it needs to be normalized immediately. It can be kept in that way except we infer some of its diagonal elements are far from being =1. We try to maintain a balance between keeping Φ close to being normalized, not necessarily being strictly normalized. Due to that, we just normalize it partially, trying to maintain its diagonal elements ϕ(i,i)≃1, but still minimizing the frequency of re-scaling (full normalizations). Still, in the GCKF framework, normalizing it after each (usually sporadic) global update is not a high processing cost, due to the low frequency nature of the GUs.

In the previous discussion, we have not described the steps for efficiently obtaining S (which is needed for obtaining the necessary component $C_{i}$), because those operations are not expensive as *m* and length(ii) are usually well lower than *n*.

## 5. Experimental Results: SLAM Case

Low precision versions of the GCKF were applied to the SLAM case, like that previously presented in [1], however in this case a mono-agent SLAM has been considered, for a more compact presentation of the results. The results from the same problem treated by the standard double precision GCKF and by the lower precision ones (in fact by multiple versions of lower precisions, to compare diverse precisions) are shown in this section.

For instance, the discrepancy between the estimates of the double precision version and those of the 24 bits version is marginal.

The covariance matrix is offered in blocks, under request from clients (in place of requiring full conversion between integer precision and nominal precision); consequently, there is an improvement in processing time, due to the way the computer memory is used. These improvements are appreciated in the times needed for processing the global updates, as shown in the processing times for the usual GCKF and its version operating in low precision floating point format, in the high dimensional global component of the GCKF. The results were produced and collected from a laptop with Intel i7-1185G7 CPU at a base frequency of 3.0 GHz and two physical 16 GB RAM.

### 5.1. Description of the SLAM Process

The SLAM process is performed in a simulated context of operation, in 2D, in which many Objects of Interests (OOIs) are well spread around the area of operation. In addition to those landmarks, walls are also present, so that occlusions to the sensors’ visibility are also included, for making the simulation realistic. These are illustrated at a certain update event in Figure 2 and Figure 3. The SLAM process is performed considering that no speed measurements are available, but the IMU gyroscope is present (although it is polluted by bias whose value is jointly estimated in the SLAM process). The observability of the full SLAM process is still achieved due to the consistent detection of a high number of landmarks. Observations are range and bearing; however, the quality of the range components is poor, in comparison to those of the bearing. Discrepancies between actual (i.e., simulated in our experiment) values and the expected values of positions are provided. Those always show proper consistency. We focused our attention on the performance of each of the suboptimal approaches being tested, comparing their results with those of the optimal full GCKF (i.e., the double precision version).

### 5.2. Results from the Normal Precision Values

The processing times required by the standard GCKF are shown in Figure 4. The GCKF is efficient in terms of processing cost; however, the global updates (i.e., those updates that affect the full global PDF, and which do occur at a low rate) result in peaks of processing; those peaks can be seen in Figure 4.

The processing times for the GCKF operating under low precision global PDF are shown in Figure 5. The regular peaks for those updates which required a global update are well less expensive than those equivalent ones in the double precision standard GCKF in Figure 4. This saving in processing time is achieved at no sacrifice in accuracy. The saving in processing times is mostly due to aspects related to memory usage (e.g., fewer RAM cache misses) than to actual processing effort. It is worth noting that both experiments are identical, even in the instance of noises.

It is interesting to note that the processing times at the non-GU (global update) instances (at which the updates just occur in the high frequency estimation processes, not involving a global update) slowly increase as the estimation keeps progressing during the SLAM process. That increase is due to the increase in the number of states, as the map’s size does increase in the exploration (as shown in Figure 6). The processing times corresponding to global update events are also affected by the increase in the number of states being estimated, as the map grows.

The average time is 24% lower than that of the full precision mode (19.5 ms against 26.5 ms). However, the relevant benefit is more related to the reduction in the cost of the sporadic global updates of the GCKF, and on reducing the memory usage. The regular peaks for those updates which require a global update are well less expensive than those equivalent ones in the full precision standard GCKF. A question would be “Where is the saving in processing time?”. A critical component in the answer would be that by reducing the required memory for the large covariance matrix and the frequency at which it is fully accessed, the number of cache misses is reduced dramatically, resulting in latencies closer to those of the static RAM of the cache memories. That is an intrinsic benefit of using the GCKF. The proposed new additional processing further improves the efficiency of the GCKF in accessing memory. It can be seen that for the standard GCKF, the peaks of processing time do happen at the times of the global updates. With the additional capability of storing the covariance matrix in a low precision numerical format, we are improving it, on average, by a 50% reduction in processing time. Each EKF update usually processed 70 scalar observations (about 35 landmarks being detected/scan). Those were grouped in sets of 15/updates, requiring multiple individual updates for completing the update. In addition, iterations were applied to treat the non-linear observation equations (a few iterations, usually 3).

The travelled path of the platform in the simulation for both the double and single precision versions of GCKF are shown in Figure 7. The maximum discrepancy between both estimation processes was about 0.3 mm, which is not significant due to the scale of tens of meters of the trip and the area of operation. Multiple loop closures is observed during the trip as the platform revisits certain areas.

The positional discrepancy between the standard GCKF with double precision and the simulated ground truth data is shown in Figure 8.

The top-left plot in Figure 9 shows the discrepancy/difference of the (x,y) position estimates of the platform between the standard double precision GCKF and those of the single precision one (in millimeters) at each update instant during the trip. The top-right plot, on the other hand, shows the differences in the standard deviations of the marginal PDFs for the estimated position. The one based on the bounded single precision covariance matrix is always slightly more conservative than the standard one. Those values are also consistent with the discrepancies in the expected values as seen in the bottom plot of the figure. This is a relevant result, as the SLAM process is usually characterized by a cumulative error as the platform travels far away, only to be mitigated by the loop closures.

### 5.3. Results of the Lower Precision Versions

This section presents the results of the approximated cases based on the lower precision numerical formats used for storing the large covariance matrix of the GCKF global component. A number of cases were considered. We have particular interest in the cases in which the global covariance matrix is approximated based on low precision such as 8 and 16 bits (results shown in Figure 10 and Figure 11). For the 8 bits cases, the GCKF estimates indicated a relevant discrepancy with the ground truth, still we include the results of that case, for showing the effects of highly aggressive approximations.

Some unusual cases, such as those of 10 bits, 20 bits, and 32 bits are shown in Figure 12, Figure 13 and Figure 14. We considered 32 bits as being unusual because we expect to use lower precisions, however, int32 is a well natural native representation. 10 and 20 bits are not native integer representations but may be of interest in certain applications.

Figure 15 jointly shows the absolute error between expected positions and actual positions of the vehicle, for the cases of double precision (standard GCKF) and lower precision versions (8, 10 and 16 bits).

## 6. Conclusions and Future Work

The proposed approach for bounding large covariance matrixes by a lower precision representation is simple and easy to apply. It is not intended to be used at high frequency but in conjunction in the GCKF framework, at low frequency, in the GCKF high dimensional component. The improvement in processing times is adequate for many applications which require real-time performance.

This modification of the GCKF is one of many more being considered by the authors. One of the next objectives is developing an asynchronous version of the GCKF, for spreading the cost of the global updates in time, making the estimation process free of those low frequency but expensive updates. The research also involves the development of a decentralized version able to operate in networked contexts. In all those variants we intend to include the required extra processing in the low frequency component of the GCKF which would henceforth result in a lower overall processing cost. Alternative approaches to bounding the full covariance matrix are also being considered. It is of interest for us to consider other strategies for reducing the processing cost related to the global updates of the GCKF, in particular those techniques related to dimensionality reduction. It is of interest for the authors to optimize the implementation for the better exploitation of GPU architecture, minimizing data transfers between main memory and GPU units, also exploiting mixed precisions operations in certain massive linear combination operations, which the high-level programming languages used in this paper do not allow.

In terms of applications, authors expect to exploit the approach estimating infinite dimensional processes, which are usually modeled as SPDE, as a continuation of their work in [2,3].

## Figures and Tables

**Figure 1 sensors-23-06406-f001:**
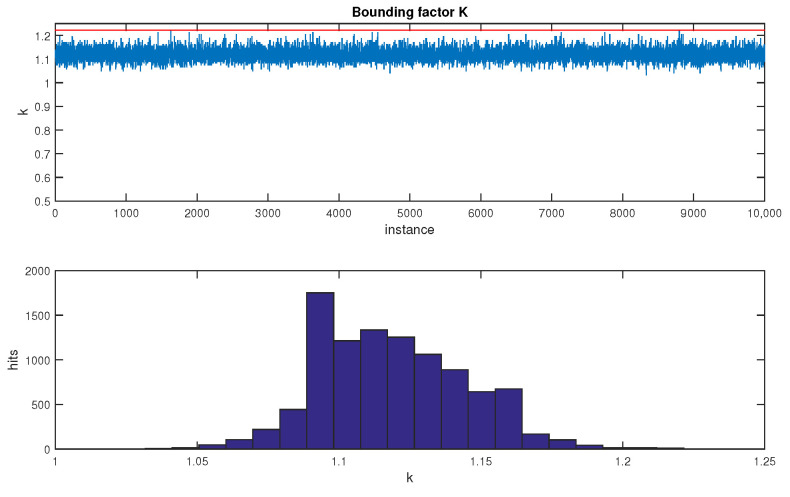
Best evaluated bounds for 10,000 cases. For each case, the bound was estimated by binary search. The maximum one was always lower than the proposed $k^{*}=1.25$.

**Figure 2 sensors-23-06406-f002:**
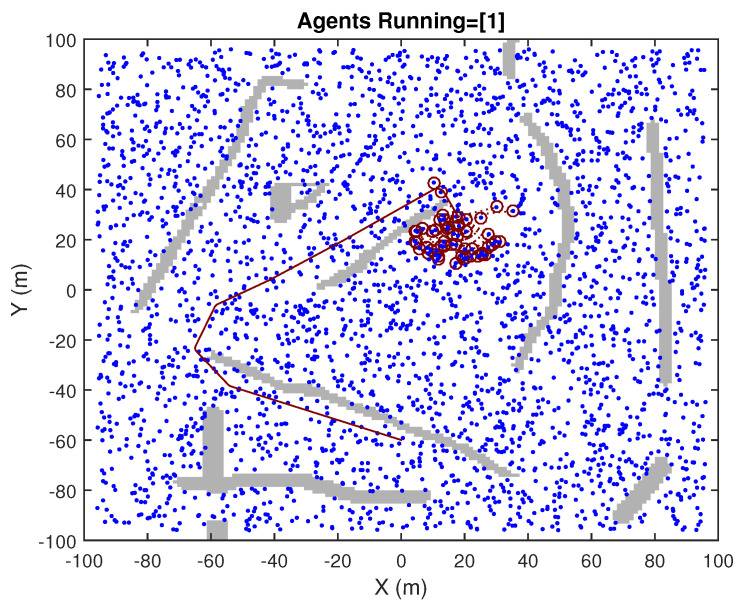
Full view, taken at a certain update event. The blue dots are OOIs (Objects of interest) on the terrain. The traveled path, until that time, is indicated by a red curve.

**Figure 3 sensors-23-06406-f003:**
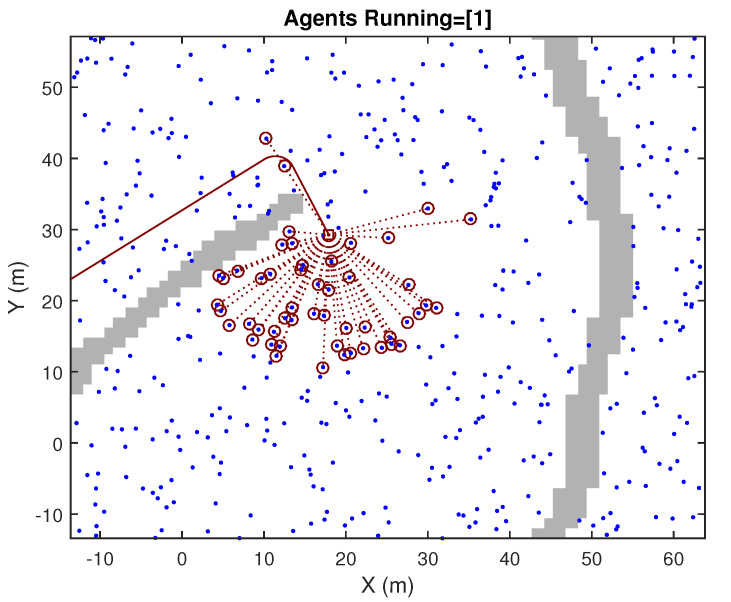
An image showing, in more detail, the OOI being detected in the previous figure (Broken red segments are used to indicate those visible OOIs.) The scanning sensor has FoV = $270^{\circ }$, but occlusions and limited range restrict visibility.

**Figure 4 sensors-23-06406-f004:**
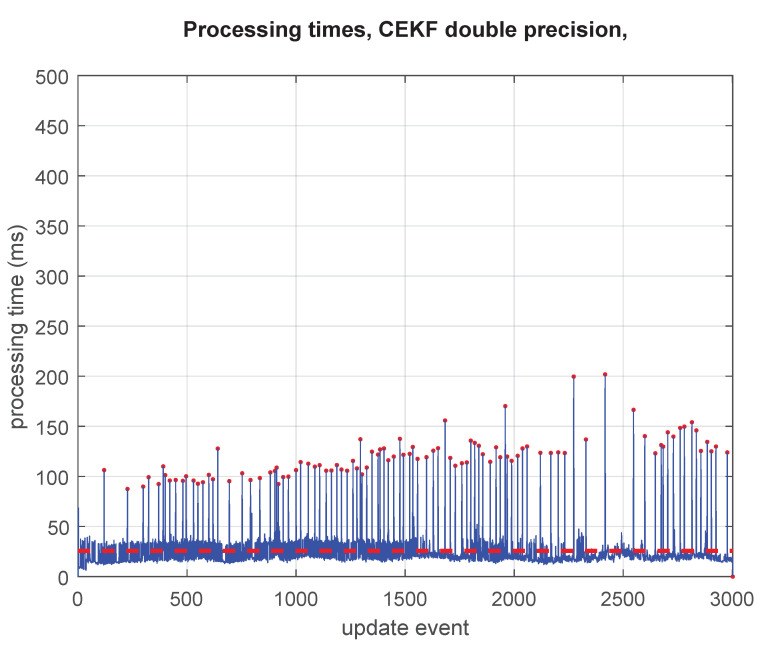
Processing times of the GCKF operating in double precision (in all its components, global low-frequency and high-frequency subsystems) at all update events are shown by the blue line. When a low-frequency global update is required, a peak in the processing time can be seen (indicated by red dots). The average processing time (24 ms), visualised by red dashed line, is well lower than that of a full filter (a standard KF implementation, which is not presented in this work), however, the peaks may represent an issue in many applications (as those peaks reached up to 200 ms with most peaks at 150 ms of processing time).

**Figure 5 sensors-23-06406-f005:**
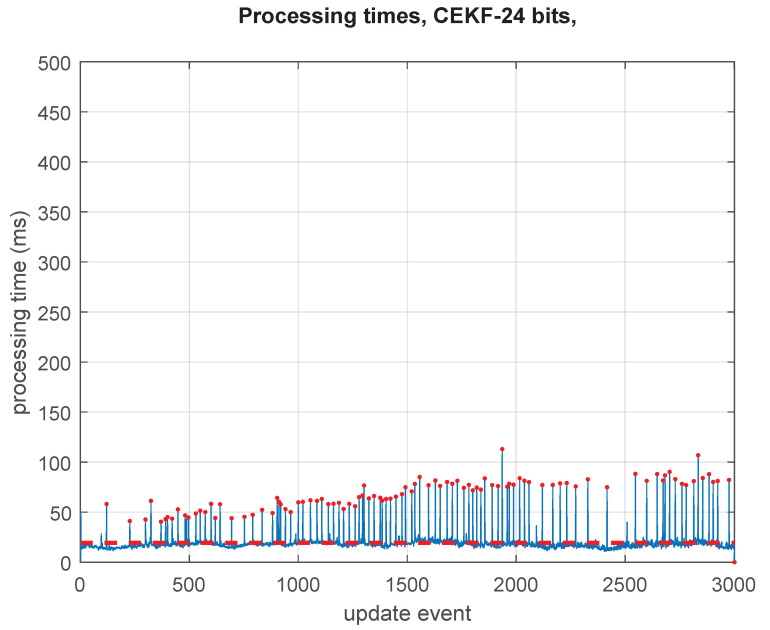
Processing times of a GCKF operating in low precision (24 bits, having its global PDF component exploiting the single precision bounding approach). The usual peaks, for those updates that require a global update, are well less expensive than those equivalent ones in the full precision standard GCKF. This saving in processing time is achieved at a negligible sacrifice in accuracy. Their implementations of the high-frequency low dimensional individual components are identical. The only difference is related to the low-frequency global component of the GCKF, which maintains the full covariance matrix in a low precision numerical format and bounds it to guarantee estimation consistency.

**Figure 6 sensors-23-06406-f006:**
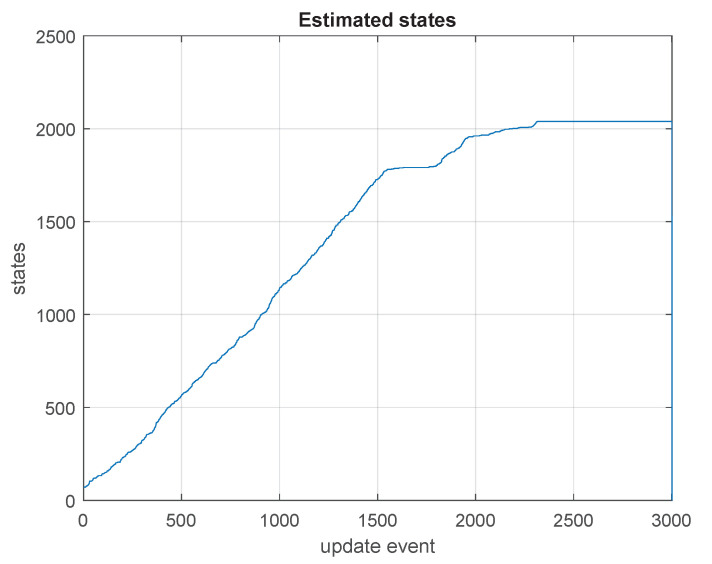
Number of states in the state vector being estimated, growing during the first phase of the SLAM process, when the map is being explored for its first time. In this part of the test, the SLAM process was in its exploration stage, and new areas were seen for the first time, making the state vector grow in length. The process reached 2 kilo-states.

**Figure 7 sensors-23-06406-f007:**
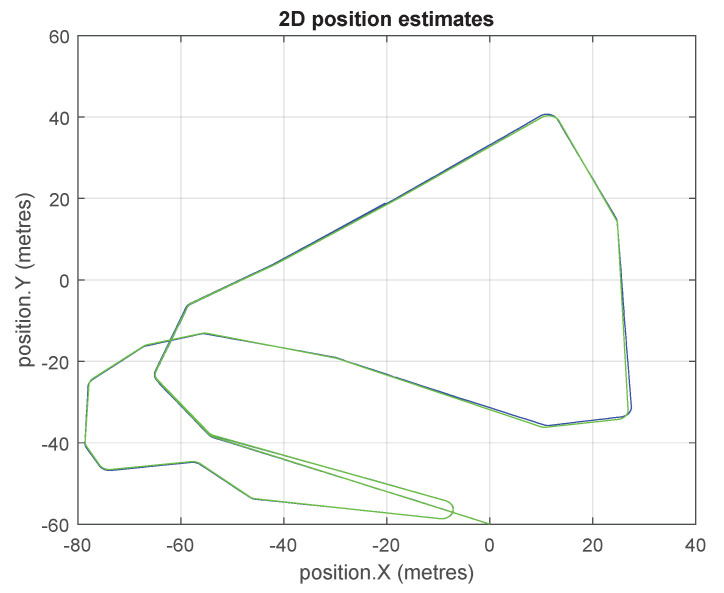
Platform’s 2D path based on position estimates, for the GCKF SLAM (double and low precision versions). The green trajectory is the actual path.

**Figure 8 sensors-23-06406-f008:**
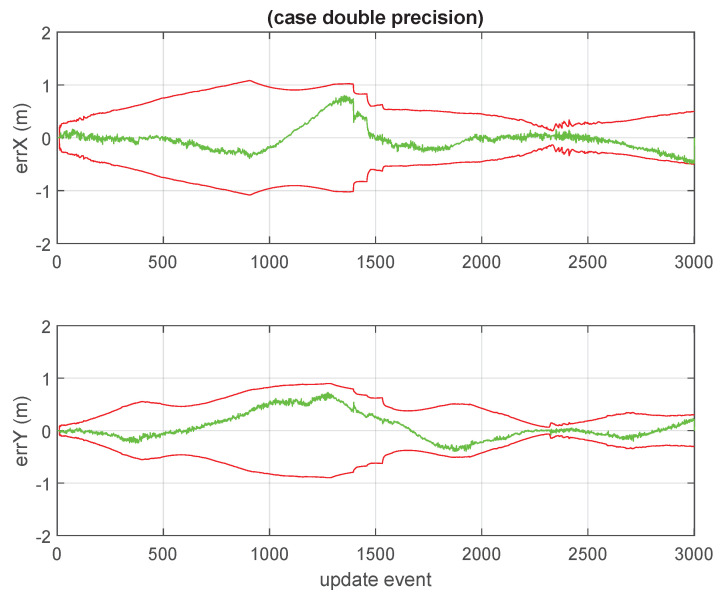
Discrepancy between ground truth and estimated expected values of the 2D position of the platform, for the standard GCKF process; those are shown in green color. The red curves are the slopes defined by the standard deviations (square roots of the variances of the marginal PDFs). The trip duration is expressed in update events. This figure is relevant for being compared with similar ones achieved by the low precision versions of the GCKF.

**Figure 9 sensors-23-06406-f009:**
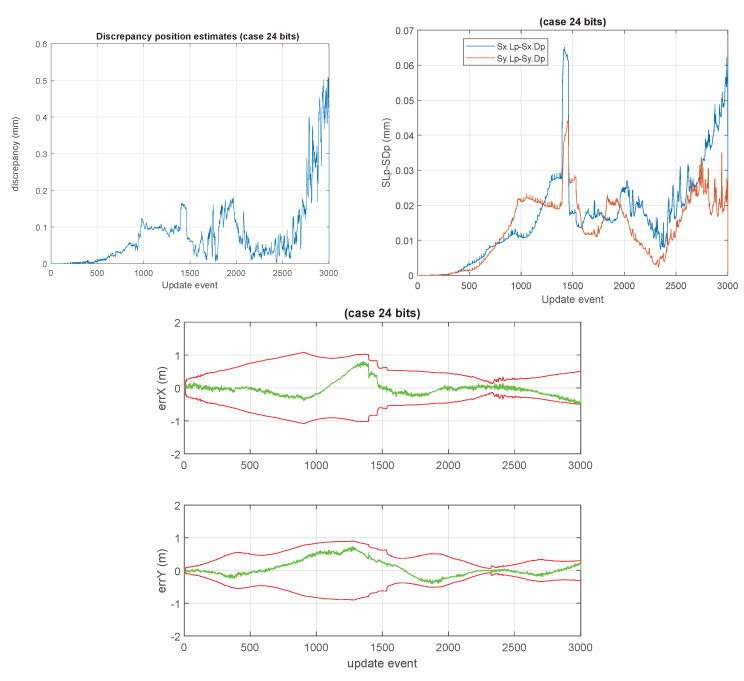
Comparing estimates of the standard GCKF and those of the “24 bits GCKF”. The first figure (top-left) shows the discrepancy between expected vehicle positions. The worst case corresponded to a distance of 0.6 mm. The second (top-right) figure shows the difference in the “marginal standard deviations” (the standard deviations of the estimates of the vehicle positions, as we are not showing here the full joint PDF of the vehicle pose and other states and parameters being estimated by the SLAM process). It can be appreciated that the standard deviations are very similar, those of the low precision GCKF being slightly higher (i.e., more conservative) than those of the standard double precision GCKF. Finally, the last figure (bottom) on the right shows the results of the 24 bits GCKF in the same way previously shown for the standard GCKF in Figure 8. The estimated states of the 24 bits GCKF are almost identical to those of the double precision GCKF.

**Figure 10 sensors-23-06406-f010:**
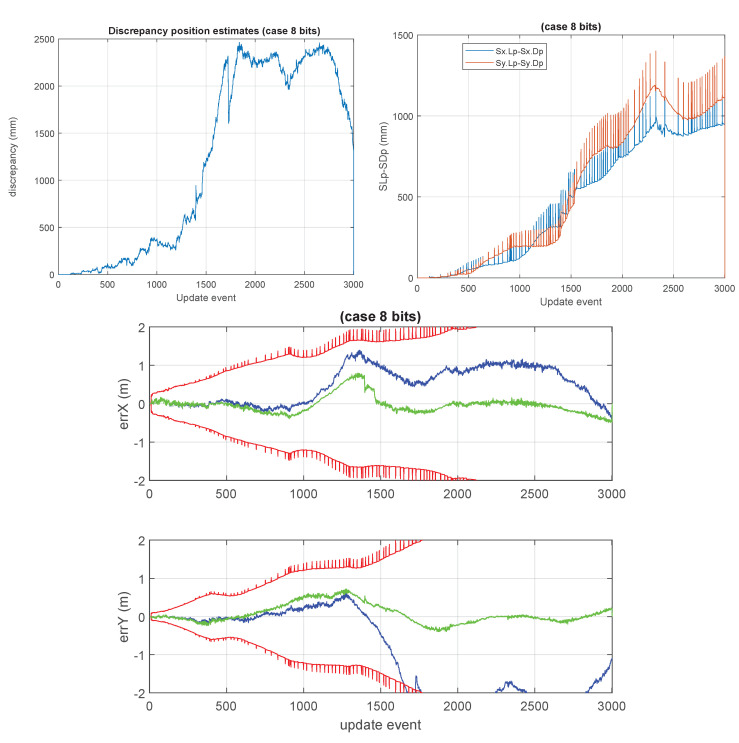
“8 bits GCKF”. 8 bits, using 7 bits for mantissa and 1 bit for its sign. The maximum discrepancy was large, about 2.5 meters. The loss of statistical dependency in the global PDF of the GCKF, due to the approximation in the covariance matrix, is too relevant, compromising the convergence of the estimation process. This SLAM process is not able to provide adequate localization accuracy.

**Figure 11 sensors-23-06406-f011:**
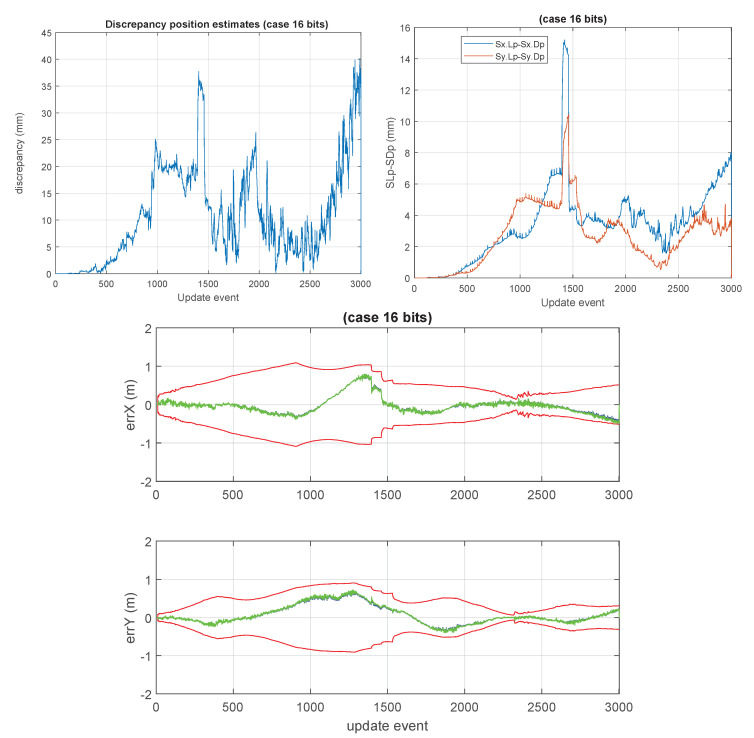
“16 bits GCKF”. Similarly, to what is shown in Figure 9, but now for the 16 bits version of the GCKF. The low precision GCKF, i.e., using 16 bits (15 bits mantissa, 1 bit its sign) for representing elements of the global covariance matrix). The maximum discrepancy was about 40 mm. The differences in the generated marginal standard deviations were consistent.

**Figure 12 sensors-23-06406-f012:**
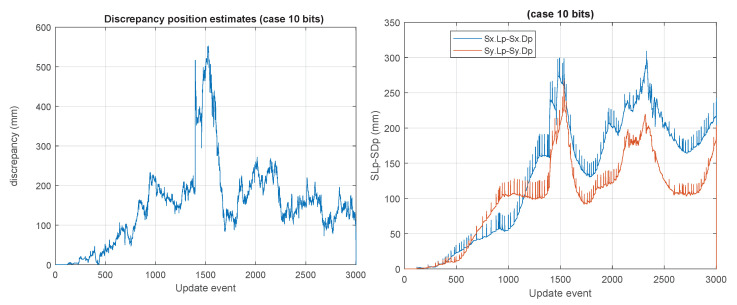
“10 bits GCKF”, whose global component maintains the 10 bits (9 bits mantissa, 1 bit its sign) bounded version of the full global covariance matrix. The maximum discrepancy was about 550 mm.

**Figure 13 sensors-23-06406-f013:**
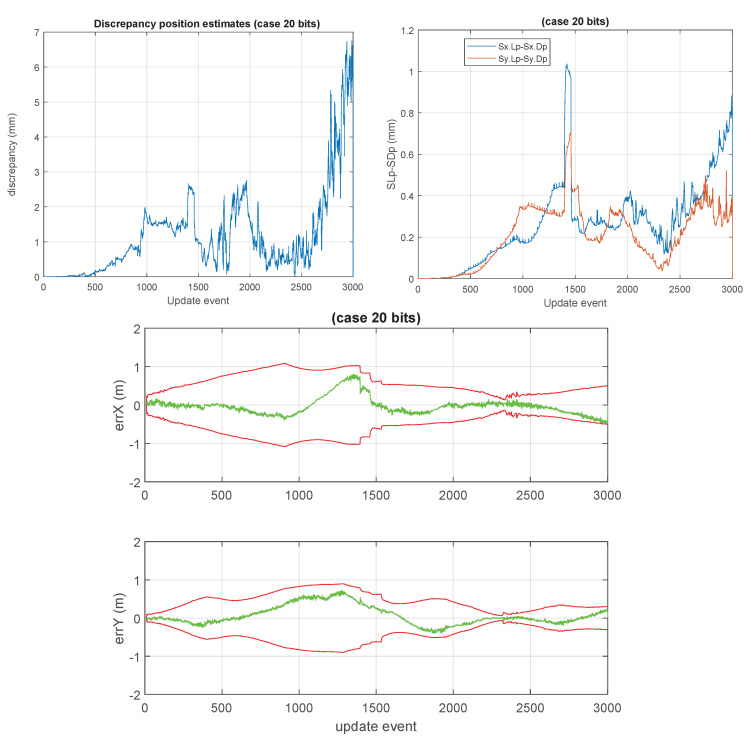
“20 bits GCKF”. Case using 20 bits (1 bit for sign, 19 bits for mantissa). Discrepancies are of a few millimeters. However, this numerical format requiring 2.5 bytes integer is not native but may be of interest for storing results.

**Figure 14 sensors-23-06406-f014:**
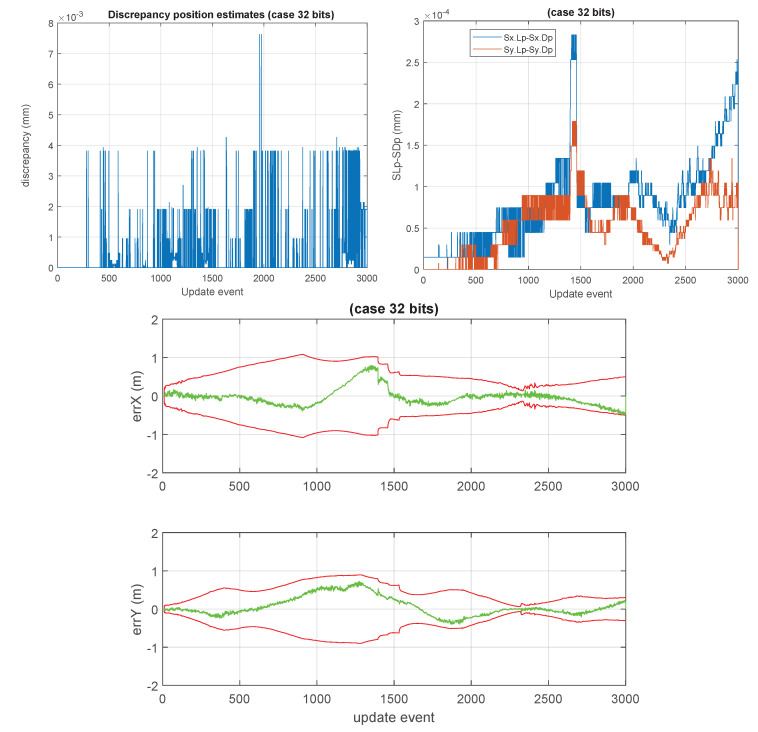
“32 bits GCKF”. Case using 32 bits (1 bit for sign, 31 for mantissa). The discrepancy related to position estimates and standard deviations are negligible, with respect to those of the double precision representation (FP64). The maximum discrepancy in the 2D position is lower than 0.008 mm. This case achieves well superior accuracy to that of the FP32 (which has only 24 bits for mantissa, in contrast to this 32-bit mantissa equivalent, achieved by a 32 bits integer representation used by the 32 “bits GCKF”). This “32 bits” case would correspond to a hypothetical FPXX, located between FP32 and FP64. This 32-bit version is an accurate, in practical terms, replacement for the double precision GCKF.

**Figure 15 sensors-23-06406-f015:**
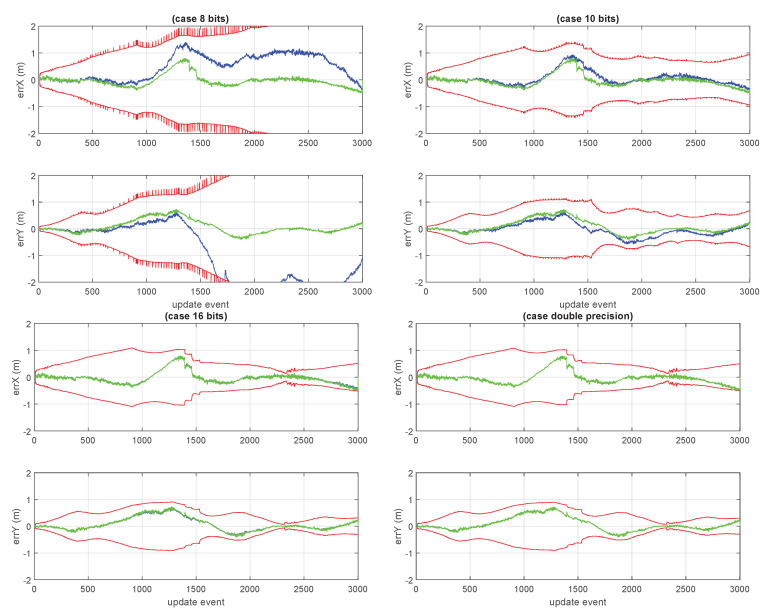
Shows the standard deviations of the vehicle’s position estimates and the associated error (of the expected values and the ground truth), for the GCKF in 8, 10, 16 bits, and for the double precision one. The sacrifice of precision for the 8 bits version, in the representation of the global covariance matrix, made the estimation process unable to converge. However, with just 16 bits, the estimation process does converge well and is close to the full precision one (not shown here, but it can be appreciated in Figure 11). Bits lengths of 16 or more bits, seemed well appropriate for this estimation problem.

## Data Availability

Not applicable.

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
