# Peer review of "Compressed Gaussian Estimation under Low Precision Numerical Representation"

_sensors, 2023, doi:10.3390/s23146406_

Round 1

Reviewer 1 Report

This paper studies the problem of compressed Gaussian estimation under low precision numerical representation. The topic is interesting.

 Comments:

The writings need to be improved to be more logical and clear. The mathematical model of the problem should be more clear.

In the introduction, the state of the art should be appended.

The description of the new algorithm is not clear. What is the new information provided by the proposed method?

Make the introduction of the proposed method in the abstract better.

What is the relationship between the proposed method and GCKF? What does combine mean?

minor editing

Author Response

Reviewer 1 

Thank you for your kind patience and critical comments. We have tried our best to clarify the issues in our manuscript according to your following comments: 

  • First, can the proposed technique be implemented in GPU? Which platforms do you consider as a potential targets for your algorithms?

Response: We usually focus on standard CPUs. For the cases of GPUs, it may also be well convenient, as new GPUs have the capability to perform linear combinations implementing accumulators which are temporarily in long precision, before producing the result. One of the authors of this paper does have experience in GPUs, and will be implementing our approach in that context, because we need to solve some large estimation problem related to 3D stochastic PDEs.   

  • “Can the proposed technique be implemented for other types of filters, e.g. filters based on delta operator for low-precision data representation?”

 Response: Our work is focussed on estimating the state of very large systems, such as those related to SLAM and more specially those related to stochastic PDEs, in which the state vector may contain thousands of states, in a native fashion (i.e. in which the high dimensionality is in the original system, and not generated to deal with augmentation for treating delays, etc). The aim is not to reduce number of operations (that is mitigated by the GCKF) but to reduce the footprint in memory usage to thus reduce the number of cache misses which introduce latencies and other effects in the CPU operation. However, the reviewer has mentioned a matter that got our attention, and we will investigate and try to understand its implications.  

  • “I recommend adding more thorough comparison between floating-point double precision, single precision and 16 bit implementation. The current version of the paper is a bit hard to perceive, despite the experiments look persuasive.”

Response: Yes, we do agree with the reviewer’s comment. Now experiments for diverse bits lengths have been added (10, 16, 20, 24, 32 bits). We are glad that the reviewer mentioned that matter, as, we think, it resulted in an interesting section of the paper. Those are implemented in the following way:   We just store unsigned int for the mantissa using 9,15, 19, 23, 31 bits, plus a bit for sign and the exponent is implicit (=0). When a section of the covariance matrix is needed in the KF update the corresponding elements are regenerated in double precision by simple assembling those in binary and latter reading those as double (that is possible because have done in a MEX function in Matlab). That can be also done using “typecast” function in Matlab.  We mention it, because there do not exist any existing native FP representation having those length of mantissas.  However, the assembling is cheap and occurs at low frequency (in the GCKF global updates). 

  • “Can the experiment shown in Fig. 5 be efficiently reproduced in 3D space?”

Response: We interpret that the reviewer is talking about the 3D SLAM implementation. In that case, our answer is yes, although for the moment we have just implemented a 2D SLAM.    

  • “Which is the minimal bit depth for using your approach? It is known, that FXP datatypes can be of arbitrary bit length, e.g. 10, 12, 14 bits etc.”

Response: That is “the question”, the reviewer is well right, we can choose arbitrary number of bits for creating our desired FP representation, even if those are not native to the CPU or GPUs used in the processing.  In the way we scale the covariance matrix we know its elements will always be in the range [-1,+1], so we know the exponent can be forced to be 0 and the mantissa used to define the FP number. We do not care if the FP number is temporarily denormalized as losing some bits for not adjusting the exponent for very small numbers is irrelevant, because the cross variances are relative to the normalized variances ( = 1), so that very small values of certain cross-variances are =0 in, practice (e.g.  a cross-variance P(i,k) = 0.0001   for variances p(i,i) = p(k,k)=1, is in practise =0 (this is valid because we use a normalized covariance matrix, and all these mentioned approximations are applied to that one, no to the actual covariance matrix). 

Our dream was to reduce the precision as much as possible, but there are limits, before we destroy the statistical dependency of the Random Variables we are modelling.. We did not know it a priori, but we knew after trying. Those limits depend on what accuracy is acceptable. We tried 8 bits, as it is a well convenient size (just one bye), but, at least  in this application, it resulted in a bias in the estimates, and the estimates were meters away from those of the “optimal” ones, provided by the double precision approach. That was not good enough. But it is good to mention the 8-bits estimator did not lie, it was still consistent in the sense it reported a covariance matrix which was consistent with the error of the expected values. However, our intention is to obtain a low precision approach whose result is well close to that of the double precision one (those are the goal, the best we would achieve). 

We were well satisfied with 20 and 24 bits. The 32 bits one was literally identical to that of the native FP64 (double), requiring just 50% of the memory. The good matter here is that the bits used for representing the numbers are fully dedicated to the mantissa (because the exponent is implicitly 0), so that those 32 bits are well more than those dedicated in a native FP32 (whose mantissa is composed by 23 bits, as the rest of the FP32 bits needs to be dedicated to the exponent + sign).  

Also, 10 bits did show reasonable accuracy, but 10 bits is not a “natural” size. Now, 12 bits sounds more natural (1byte + 1 nibble), and off course, 16 bits is good in both senses. 

24 bits is pretty accurate (performed as good as the full double precision, requiring just 24 bits in place of 64 bits of the native double precision format.  We will see when applied to SPDEs, what are the results. We tried in some cases, and the results were also very good. 

  • “What about scaling procedures? Can some additional round-off tricks be applied to reduce the impact of digital noise?

Response: That would be great. How to increase efficiency when using the same number of bits, to reduce bit lengths even more or to reduce the effects of truncation for the same number of bits. 

We are not sure about other tricks and manipulations to minimize the effects of the limited precision. Maybe some non-linear mapping to use bits more efficiently when representing those normalized cross-variances.  The question from the reviewer is trigging in us a good interest about that matter. Thank you. 

  • Can vector and matrix-support processor improve the performance of the algorithm further?

Response: Yes, we agree. Exploiting certain aspects of the Single instruction, multiple data (SIMD) capabilities of the new CPUs. Those are already used in high level languages such as Matlab and its built-in functions. Currently, we have not been digging into that and relied on Matlab doing its part. We found limitations on that “high level” usage, but we compensated those because the effect of reducing memory was well relevant as the reduction in cache misses in current computers is still a relevant matter. 

  • In my opinion, the experimental part of the study should be a bit clarified, including the purpose, targets and goals of experimental work. I also recommend expanding the description of used software to increase repeatability.

Response: We have added details in that respect. We think a good point is showing how the usage of certain matrixes in the Kalman update were organized to obtain double precision performance without converting the full covariance matrix to double.  

  • Nevertheless, my overall impression is fine and I can recommend the reviewed manuscript for publication after only minor revisions.

Response: Thank you, we feel well glad about the opinion, and about the constructive comments, which have been considered in the new version of the paper. Thank you. 

Reviewer 2 Report

The paper is devoted to the low-bit implementation of optimal Gaussian estimation algorithm by bounding large covariance matrixes which is of certain interest for embedded applications. The Authors estimate the approximation numerical error to show that covariance inflation is minimal. Developing hardware-targeted algorithms for embedded compressed sensing systems is relevant because the direct implementations of floating-point methods are not always possible even in modern embedded computers. Besides, I have several remarks regarding this paper.

First, can the proposed technique be implemented in GPU? Which platforms do you consider as a potential targets for your algorithms? The performance of short-precision

Can the proposed technique be implemented for other types of filters, e.g. filters based on delta operator for low-precision data representation?

I recommend adding more thorough comparison between floating-point double precision, single precision and 16 bit implementation. The current version of the paper is a bit hard to perceive, despite the experiments look persuasive.

Can the experiment shown in Fig. 5 be efficiently reproduced in 3D space?

Which is the minimal bit depth for using your approach? It is known, that FXP datatypes can be of arbitrary bit length, e.g. 10, 12, 14 bits etc.

What about scaling procedures? Can some additional round-off tricks be applied to reduce the impact of digital noise?

Can vector and matrix-support processor improve the performance of the algorithm further?

In my opinion, the experimental part of the study should be a bit clarified, including the purpose, targets and goals of experimental work. I also recommend expanding the description of used software to increase repeatability.

Nevertheless, my overall impression is fine and I can recommend the reviewed manuscript for publication after only minor revisions.

The quality of English language, grammar and style is satisfactory.

Author Response

Thank you for your kind patience and critical comments. We have tried our best to clarify the issues in our manuscript according to your following comments: 

The article entitled “Compressed Gaussian Estimation under Low Precision Numerical Representation” is well-written and, from my point of view, would be of interest for the readers of Sensors. In spite of this, and before its publication, the following changes should be performed: 

  • Abstract: it seems too long. Please, check if the length does not exceed the máximum of 250 words.

Response: Thank you for the comment. We have modified the abstract as per the suggestions of the reviewer in the revised manuscript.

  • “Line 47 please give any example, or at least a reference of the numerical problems that would be found.”

Response: The problems about using low precision in KF applications do mostly appear in the updates of the covariance matrix. One of the reasons is that the covariance update requires a series of calculations involving massive linear combinations and other linear algebra related operations, for which the limited precision of the numerical representation is prone to introduce errors in the results. But, if those errors are introduced in the elements of the covariance matrix, it may result in overconfident covariances, or even turn to generate covariance matrixes that are not symmetric or even not positive semidefinite, making the estimator unstable an inconsistent. For avoiding that, an easy approach is using high precision floating point representation, such as double (64 bits). Using numerical precision is usual in many applications but using it for very large covariance matrixes may be too expensive and make the CPU to operate inefficiently. 

  • The introduction requires of a most precise description of the objective and, also, of a paragraph with the summarizes the sections of the manuscript.

Response: Thank you for the comment. We have modified the introduction as per the suggestions of the reviewer in the revised manuscript. 

  • From my point of view, there is a lack of bibliographical references in section 3. Approximating low precision covariance, starting from line 121 when the first formula is presented.

Response: Thank you for the comment. We have added some updated related works as per the suggestions of the reviewer in the revised manuscript. 

  • About the title “3.1 What happens in the unlikely cases in which the bounding matrix is not true?”. Fromm y point of view this is not formal enough for a scientific journal. Please, change it.

Response: Yes. We agree. As our bounding approach is statistical, not a deterministic one, we wanted to mention that matter, in the sense that readers may ask “is this a general bounding approach?”. Well, the answer is that the bounding approach is pretty good for truncation errors such as those of finite precision. In addition, the relaxation factor “5/4(sqrt(n)” is conservative enough to make very low the probability of the bounding to be untrue. And we know that even in the case in which the proposed bounding was not true its effect would be negligible as well. However, we can even increase that factor by a 10%, thus making the probability of the bounding not being true negligeable.   

  • “Figure 5 and 6: there is a text that indicates that the figure is paused. Is it necessary? Could it be removed?”  

Response: Yes, we agree it was not good to have it there. That was the message shown in the simulation when it was paused.  So, now when the figure was saved, and later used, we removed those messages and the buttons of the GUI. 

Reviewer 3 Report

The article entitled “Compressed Gaussian Estimation under Low Precision Numerical Representation” is well-written and, from my point of view, would be of interest for the readers of Sensors. In spite of this, and before its publication, the follwing changes should be performed:

Abstract: it seems too long. Please, check if the length does not exceed the máximum of 250 words.

Line 47 please give any example, or at least a reference of the numerical problems that would be found.

The introduction requires of a most precise description of the objectived and, also, of a paragraph with the summarizes the sections of the manuscript.

From my point of view, there is a lack of bibliographical references in section 3. Approximating low precision covariance, starting from line 121 when the first formula is presented.

About the title “3.1 What happens in the unlikely cases in which the bounding matrix is not true?”. Fromm y point of view this is not formal enough for a scientific journal. Please, change it.

Figure 5 and 6: there is a text that indicates that the figure is paused. Is it necessary? Could it be removed?

Author Response

Thank you for your kind patience and critical comments. We have tried our best to clarify the issues in our manuscript according to your following comments: 

  • The writings need to be improved to be more logical and clear. The mathematical model of the problem should be more clear.

Response: We have mentioned more details about how to manipulate the large matrixes involved in the KF update, particularly describing it using Matlab like notation (and using indexes) , as it give better idea about how to operate with the matrix operations in the KF update steps. By using indexes, the usual sparse nature of some matrixes involved in those operations can be exploited. A good example is the H matrix (observation matrix), which is usually well sparse. For that reason we indicated and applied it via indexes, e.g. H(:, ii) * P(ii,: ) , in pace of a “brute force” calculation H*P. This is critical in our internal calculations.  

  • In the introduction, the state of the art should be appended.

Response: We think we have now included a better introduction and a description the state of the art in high dimensional estimation and in respect of the approaches for treating limited numerical precision (in particular for large systems)  

  • The description of the new algorithm is not clear. What is the new information provided by the proposed method?

 Response: The paper proposes a simple approach for bounding covariances matrixes respecting the constraint that the bounding matrix is defined by a matrix whose elements can be represented in low precision numerical format. The resulting bounding matrix is guaranteed to be valid by slightly/adequately relaxing its diagonal elements (inflation). The inflation effect is not high, but still, if it was applied in a normal Kalman filter at high rate, it would still make the inflation effect to grow excessively. But then we found it does not need to be applied at high rate but just at low rate in the global updates of the approach named Generalized Compressed Kalman Filter (GCKF). Because of that, the cumulated inflation of the covariance matrix is low. This is possible because we use the bounding approach just in the Global updates (GU) of the GCKF. We consider it is a good combination of our GCKF approach and the efficient bounding approach. 

  • Make the introduction of the proposed method in the abstract better.

Response: Thank you for the comment. We have modified the abstract and introduction as per the suggestions of the reviewer in the revised manuscript.

  • What is the relationship between the proposed method and GCKF? What does combine mean?

Response: Yes. The GCKF is an approach for treating the estimation of the state vector of very large systems (e.g. hundreds to thousands of states), such as those related to Stochastic PDEs, and also Bayesian SLAM.  The GCKF divides the problem into multiple smaller estimation processes (i.e. of lower dimension) which do operate at high rate (at the rate that would require a standard KF). In addition to those multiple small estimators, the GCKF performs certain updates at low processing rate, but those are of very high dimension (that of the full state vector being estimated). Those updates are named “global updates”, (GU). Each time a GU is performed, a large covariance matrix needs to be updated, and that step is expensive (but, still, it is a good deal, because in the overall estimation process it is well cheaper than using a normal Kalman Filter). However, the question that this paper tries to answer is: “can we make the overall GCKF even more efficient?”. The answer is “yes, we may make the GUs cheaper, by reducing the numerical precision used in storing the full covariance matrix. But why reducing the precision may help in the processing cost of the GU, if the number of operations would be the same?  The answer is that by reducing the memory usage for the large full covariance matrix the latencies are dramatically reduced, because the number of memory cache misses are dramatically reduced, and so the processing times is reduced as well.  Consequently, the proposed method has the merit of contributing with that improvement in performance. 

Round 2

Reviewer 1 Report

No more comments